# Genetic Drivers of Head and Neck Squamous Cell Carcinoma: Aberrant Splicing Events, Mutational Burden, HPV Infection and Future Targets

**DOI:** 10.3390/genes12030422

**Published:** 2021-03-15

**Authors:** Zodwa Dlamini, Mohammed Alaouna, Sikhumbuzo Mbatha, Ahmed Bhayat, Mzubanzi Mabongo, Aristotelis Chatziioannou, Rodney Hull

**Affiliations:** 1SA-MRC/UP Precision Prevention & Novel Drug Targets for HIV-Associated Cancers Extramural Unit, Pan African Cancer Research Institute, University of Pretoria, Hatfield, Pretoria 0028, South Africa; achatzi@bioacademy.gr (A.C.); Rodney.hull@up.ac.za (R.H.); 2Department of Internal Medicine, Faculty of Health Sciences, University of the Witwatersrand, Johannesburg 2050, South Africa; 321814@students.wits.ac.za; 3Department of Surgery, Steve Biko Academic Hospital, University of Pretoria, Hatfield, Pretoria 0028, South Africa; sikhumbuzo.mbatha@up.ac.za; 4Department of Community Dentistry, School of Dentistry, University of Pretoria, Hatfield, Pretoria 0028, South Africa; ahmed.bhayat@up.ac.za; 5Department of Maxillofacial and Oral Surgery, School of Dentistry, University of Pretoria, Hatfield, Pretoria 0028, South Africa; mzubanzi.mabongo@up.ac.za; 6e-NIOS Applications PC, Kallithea, 17676 Athens, Greece; 7Center of Systems Biology, Biomedical Research Foundation of the Academy of Athens, 11527 Athens, Greece

**Keywords:** head and neck squamous cell carcinoma (HNSCC), aberrant splicing events, human papillomavirus (HPV) infection, non-coding RNA (ncRNA), methylation, mutational burden

## Abstract

Head and neck cancers include cancers that originate from a variety of locations. These include the mouth, nasal cavity, throat, sinuses, and salivary glands. These cancers are the sixth most diagnosed cancers worldwide. Due to the tissues they arise from, they are collectively named head and neck squamous cell carcinomas (HNSCC). The most important risk factors for head and neck cancers are infection with human papillomavirus (HPV), tobacco use and alcohol consumption. The genetic basis behind the development and progression of HNSCC includes aberrant non-coding RNA levels. However, one of the most important differences between healthy tissue and HNSCC tissue is changes in the alternative splicing of genes that play a vital role in processes that can be described as the hallmarks of cancer. These changes in the expression profile of alternately spliced mRNA give rise to various protein isoforms. These protein isoforms, alternate methylation of proteins, and changes in the transcription of non-coding RNAs (ncRNA) can be used as diagnostic or prognostic markers and as targets for the development of new therapeutic agents. This review aims to describe changes in alternative splicing and ncRNA patterns that contribute to the development and progression of HNSCC. It will also review the use of the changes in gene expression as biomarkers or as the basis for the development of new therapies.

## 1. Introduction

The term head and neck cancers are used to describe a variety of tumors that arise in the mouth, nose, throat, sinuses or salivary glands [1]. Head and neck cancers are the sixth most common form of malignancy, with a total of 600,000 reported cases around the globe each year [2]. Over 90% of these cases are squamous carcinoma of the head and neck, head and neck squamous cell carcinoma (HNSCC) [3]. More than two-thirds of HNSCC incidents are diagnosed in developing countries [4]. The estimated average age of patients is 60 years, and the incident rate is highest in males [5]. The first indications that a patient is suffering from these types of cancers include changes in the sound of the voice, a persistent sore throat that will not heal, difficulty in swallowing and most notably, the development of lumps or lesions in the throat [6]. Even with major advances in diagnosis, radiation therapy and immunotherapy, the 5-year survival rate for HNSCC patients has not improved in recent decades [7,8]. Additionally, due to the lack of appropriate biomarkers for the early diagnosis of HNSCC, in many patients, the cancer is only detected at the later stages of the disease, leading to a poor prognosis [4,9].

The primary risk factors for HNSCC involve smoking and heavy alcohol use [10]. Human papillomavirus (HPV) is classified as a distinct risk factor, giving rise to tumors that are distinct from those caused by other risk factors [11]. Genome-wide systematic sequencing of mRNAs, microRNAs (miRNAs), long non-coding RNAs (lncRNAs), and circular RNAs have led to the identification of probable methylation sites, single nucleotide polymorphisms (SNPs), mutations and variations in copy number in a variety of different genres. This has led to the identification of numerous potential biomarkers for HNSCC [12,13,14,15,16]. In addition to these genomic and epigenetic changes, alternative splicing events have also been implicated in the initiation and progression of head and neck cancer [17].

## 2. The Role Played by HPV Infection in HNSCC Development and Progression

Another important factor for the changes in gene expression that occur in head and neck cancer is infection with HPV. This is an independent etiological factor in the development of HNSCC and has been the target of interest for a large amount of recent research [11,18,19]. Human papillomaviruses are epitheliotropic DNA viruses with an average genome size of 8 kb [20]. The virus generates two oncoproteins encoded by the E6 and E7 genes that effectively inhibit the proteins p53 and pRb. This leads to the initiation of the cell cycle and DNA synthesis, which is required for viral replication [20]. Numerous studies have shown that HPV positive (HPV+) and HPV negative (HPV-) HNSCCs are separate entities with distinct etiologies, clinical behaviors, treatment outcomes, pathological toxicity, and molecular profiles [21,22,23,24]. HPV type 16 is identified as the causative agent in more than 90% of HNSCC cases [25].

Tumors expressing HPV genes (particularly HPV16) displayed no TP53 mutations and low losses of segments of chromosomes 3p, 9p and 17p [19]. Traditionally HPV+ HNSCC is more sensitive to treatment; however, resistance to these treatments, such as chemotherapy, radiation, and surgery, are on the rise [26]. Most HPV+ HNSCC tumors were found to be infected with HPV16 alone and showed expression of the HPV16-E6 oncogene [26]. A distinct class of 123 member genes was specifically deregulated in HPV16 positive HNSCC. These genes were deregulated in both smokers and non-smokers [26]. The symptoms of HPV+ and HPV- HNSCC are very different from each other, which causes confusion about whether these cancers are considered distinct tumors [27]. HPV + oral cancers show changes in the expression of genes regulating the cell cycle or a decrease in the levels of tumor suppressor proteins, such as pRb and cyclin D1. These proteins are usually overexpressed in oral HPV-e tumors [27,28].

### HPV-Infected HNSCC Expression Profiles

HPV + HNSCC tumors overexpress retinoblastoma-binding protein factor-C replication gene, and transcription factor partner E2F-dimerization protein (TFDP2). A large group of genes that play a role in the defense against viral infection and immune response have been shown to be ineffective against HPV, including interleukin and interferon-induced proteins [26].

A study has demonstrated that there is a difference in the expression pattern of host genes in HPV + tumors from smokers, ex-smokers and non–smokers. HPV16-+ tumors from smokers could be monitored through the expression of p53 or E2F-transcription factors, such as insulin growth factor (IGF), protein transcription factor-C4 (RFC4), cell division cycle (CDC7), cytochrome P450 (CYP4V2), mini-chromosome maintenance protein complex 2 (MCM 2) and mitotic checkpoint complex protein (MCC) [29,30]. It was proposed that the expression of some of these genes might be linked to tobacco use [26]. Cyclin-dependent kinase inhibitor 2C (CDKN2C) and retinoblastoma (RB) genes were among the genes whose expression is consistently and greatly altered in HPV16-positive tumors from non–smokers. Neither of these genes has been conclusively linked with HPV16 in smokers despite their upregulation [26]. CDKN2C encodes the enzyme p18, a tumor suppressor and cyclin-dependent kinase receptor. It binds protein kinases and acts in conjunction with the retinoblastoma tumor protein (pRb) to inhibit cell cycle progression and regulate growth [26]. Enhanced CDKN2C and RB expression; suggests the lack of the negative feedback loop, a situation that is observed when the expression of HPV16-E7 is repressed [22]. Studies found that the expression of CDKN2 was able to regulate the growth of cell lines derived from HPV16-+ HNSCC tumors [31]. Cancer development and progression can also be inhibited in HPV-infected cells through the ability of pRb to suppresses the function of the E2F transcription factors in cells infected with HPV [32,33].

In head and neck cancer, p16 is inactivated by gene mutation or methylation, which triggers the functional inactivation of pRb [26,34]. HPV-E6/E7-related overexpression of p16 protein occurs in oral lesions [35]. In HPV16-+ HNSCC, elevated levels of the expression of the cell cycle control genes (CDC7, MCM2) were also recorded [22,31]. Regulation of interferon-inducible protein (IFN-inducible) and interleukin-1 receptor antagonist (IL-1RA) was recorded in-HPV16 expressing immortalized head and neck tumor cell lines [36]. Spontaneous degradation of the early HPV protein E2 led to increased transcription of viral DNA. This is accompanied by the rise of antiviral gene expression in the form of IFN and an increase in viral E6/E7 oncoprotein production [37]. Other studies also demonstrated that transfection of malignant cells with oncogene E7 would render them more vulnerable to IFN-alpha-induced apoptosis [38]. This suggests that active chronic oncogenic HPV infection can impact the vulnerability of cells to IFN-induced apoptosis in tumor tissue and may similarly affect IFN-based HPV therapy in associated diseases, such as HNSCC [34].

## 3. Alternative Splicing in HNSCC

A study published in 2019 reported the occurrence of alternative splicing (AS) events in 519 HNSCC patients. It was found that in these 519 samples, there were 4626 AS-related survival events in 3280 genes. These changes in AS signatures resulted in multiple, cumulative survival outcomes [39]. A study in 2020 identified 4068 splicing events associated with changes in the survival of HNSCC patients, using records from The Cancer Genome Atlas (TCGA). These results imply that a patient’s AS signature can be used as a prognostic biomarker [17]. The top five AS events that correlated with survival were exon skipping, use of alternate promoter sites, use of alternate terminator sites, use of alternate acceptor sites, and use of alternate donor sites (Figure 1). These results imply that AS events are capable of being not only diagnostic and prognostic biomarkers but also therapeutic targets for the treatment of HNSCC patients. GO and KEGG analysis indicates that most genes whose splicing is altered in HNSCC are implicated in playing a role in functions, such as apoptosis, DNA repair, mRNA splicing and metabolism [39].

Further analysis indicated that HNSCC patient survival was associated with AS of five specific genes. These five genes are C5orf30, eEF1A lysine and N-terminal methyltransferase (*METTL13*), Ras homolog gene family member T1 (*RHOT1*), ATP-binding cassette sub-family C member 5 (*ABCC5*), and Myelin protein zero-like protein 1 (*MPZL1*). The role played by AS in these five genes is currently not fully known. METTL13 di-methylates eukaryotic elongation factor 1A (eEF1A), leading to increased translation and protein expression and can promote cancer formation and progression [41]. METL13 is alternately spliced to give rise to 5 isoforms. The full-length protein contains two methyltransferase domains. The second, of which is missing in at least two of the isoforms. These isoforms would then be less efficient at methylating targets, implying that these isoforms could help prevent cancer formation and progression (Figure 2A). RHOT1 is a membrane receptor that promotes proliferation and cancer [42]. It is spliced to give rise to six isoforms. Some of these isoforms lack the transmembrane receptor, implying that these isoforms can block signaling and could, therefore, prevent cell migration. (Figure 2B) MPZL1 activates Src kinases, which results in increased cancer cell proliferation and migration [43]. The full-length variant has two transmembrane domains. Two of the splice variants lack one of these domains. This may interfere with the recognition of ligands by this receptor (Figure 2C). In addition to these genes, there are a variety of genes that have been found to be alternately spliced in HNSCC. Examples of these genes are discussed in more detail below.

### 3.1. DOCK5

Dedicator of cytokinesis 5 (DOCK5) is an intracellular signaling protein that is alternately spliced to give rise to at least two isoforms. By analyzing the expression changes of different isoforms of dedicator of cytokinesis 5 (DOCK5) and comparing this to clinical parameters, a link was discovered between the expression of certain DOCK5 variants and the patient’s tobacco usage. This indicates that smoking decreases the overall survival of a patient through the alteration of the expression of DOCK5 variants [44]. AS of the DOCK5 mRNA gives rise to two splice variants. One variant contains an exon with an alternate terminator site, resulting in a truncated variant of DOCK5. This variant of DOCK5 enabled HNSCC cell proliferation, migration, and invasion of HPV-negative HNSCC [44]. The DOCK family of proteins are members of the guanine nucleotide exchange factor (GEF) group. These contain two DOCK homology (DHR) domains, DHR-1, and DHR-2, where DHR-2 is the GEF catalytic element [45]. AS events in DOCK5 observed in HNSCC results in the loss of this catalytic domain [44]. The expression of a truncated variant lacking the catalytic domain can promote the development and progression of HPV-negative HNSCC [44]. This also implies that this splice variant, as well as the processes it is associated with, can serve as possible therapeutic targets.

### 3.2. Lysyl Oxidase (LOXL2) Facilitates the Development of HPV-Negative HNSCC

Lysyl oxidases (LOXL) are a family of copper-containing amine oxidases that catalyze the deamination of lysine residues in collagen and/or elastin. These lysines are involved in the formation of crosslinking in the extracellular matrix leading to increased fibroblast growth and adhesion. In this way, the overexpression of members of the LOXL family promotes metastasis [46]. There are four members of the LOXL family, and these are named LOXL1-4 [47,48]. One of these family members, LOXL2, contributes to the initiation and development of tumors [49]. LOXL2 overexpression seems to enhance the ability of cancer cells to invade tissue layers and promote metastasis [49,50]. LOXL2 activity can be inhibited by hypoxia and hydrogen peroxide (Figure 3A).

Several splice variants of LOXL2 have been reported in various cancers, including esophageal squamous cell carcinoma. Two isoforms—the exon 13-free form (Figure 3B) and the form containing a 72-nucleotide-deletion—result in tumor progression through a new molecular mechanism distinct from the canonical model for LOXL2 [51,52]. This alternate LOX2 mechanism involves the activation of signaling pathways, such as focal adhesion kinase (FAK) and protein kinase B (PKB), and leads to the transition from epithelial to mesenchymal tissue [53]. Overexpression of the isoform lacking exon 13 activated these pathways to a higher degree in HPV-negative HNSCC patients; this includes both the focal adhesion kinase (FAK) and protein kinase B (PKB) pathways. This is associated with modifications in p-FAK, p-AKTT308, p-AKTS473 and p-S6 (Phospho-S6 Ribosomal Protein) [54]. Therefore, it is predicted that this LOXL2 splice variant may promote the proliferation, migration, and invasion of HPV-negative HNSCC cells [54].

All these findings promote the idea that LOXL2 isoforms can be used as biomarkers or therapeutic targets. Its use as a biomarker is further promoted by the fact that it is secreted [55] and can, therefore, be used as a biomarker in non-invasive liquid biopsies.

### 3.3. Transcription Factor Dp-2 (TFDP2)

The transcription factor Dp-2, also known as the E2F dimerization partner 2 (*TFDP2)* gene, encodes a member of a family of transcription factors that heterodimerize with E2F proteins to enhance their DNA-binding activity and promote transcription of E2F target genes. The expression of TFDP2 is upregulated in HPV16-positive HNSCC tumors from non-smokers [26]. Some of these E2F target genes function to control the transcriptional activity of numerous genes involved in the progression from the G1 to the S phase of the cell cycle. TFDP2 is alternately spliced to give rise to 8 isoforms (Figure 4). One of these isoforms lacks the E2F transcription factor dimerization partner domain. This DNA-binding domain stimulates E2F transcription. Increased expression of this isoform may inhibit cell cycle progression (Figure 4).

### 3.4. Splicing of p53 in HNSCC

The abolition of normal p53 function is one of the most common genetic changes in human cancer. P53 mutations are assumed to contribute significantly to the development of around 40% of HNSCCs [11,56]. AS of p53 gives rise to at least 12 isoforms. These isoforms all retain the mutation hotspot sequence (exons 5–8). The canonical p53 protein (Figure 5B) is named p53α and is normally the most abundant isoform and contains all seven functional domains. The two N-terminal transactivation domains, a proline-rich domain (PXXP), a DNA-binding domain, an oligomerization domain (OD), a nuclear localization signaling domain and a negative-regulation domain [57]). The isoforms are divided into three main groups variants, α, β or γ, based on the splicing changes at the N terminal. Isoforms in group alpha have the N terminal basic domain. Isoforms in groups beta and gamma lack this domain and use an alternate exon nine splice variants, exon 9a in group beta and exon 9b in group gamma (Figure 5) [58]. The other isoforms are the result of a variety of AS mechanisms, including alternative promoter usage and alternative initiation of translation sequences. These groups each contain their own truncated variants that arise due to internal promoters Δ40p53. Δ133p53 and Δ160p53 [57].

The isoform detected at the highest level in HNSCC was p53β [58]. Unlike the canonical p53, p53β preferentially binds to the promoter for the proapoptotic *Bax* and is unable to efficiently induce the expression of the p53 regulator MDM2. This allows it to induce apoptosis in a p53 independent manner [59]. Overexpressed p53β cooperates with full-length p53 and contributes to cellular senescence. This increase in p53β was observed in vivo in senescent colon adenomas [58]. The truncated Δ40p53 isoforms are created by AS in intron 2, and the resulting isoform lacks the transactivation domain. These isoforms have a dominant-negative effect on the activity of full-length p53 [60]

The shorter isoforms Δ133p53 and Δ160p53 lack the transactivation domain, the proline-rich domain and a part of the DNA-binding domain (Figure 5). These isoforms can interact directly with the canonical p53 and regulate its transcriptional activity [58]. A mouse model was developed expressing a p53 protein with a deletion of the first 122 amino acids. This model was used to study the role of the Δ133p53 isoforms [61]. This p53 promoted hyperproliferation in cancer and inflammation in the mice studied [61]. Other studies undertaken using mice xenograft models have shown that Δ133p53α could stimulate cell migration and angiogenesis [62]. The Δ133p53α isoform is also able to prevent p53-mediated replicative senescence, G_1_ cell-cycle arrest and apoptosis [63]. In summary, p53β promotes replicative senescence, and the action of this isoform is opposite to that of Δ133p53α, which promotes proliferation. Therefore, the ratio of p53β/Δ133p53α can be used to measure cancer risk. A decrease in the ratio would favor cancer development and progression [58].

**Figure 5 genes-12-00422-f005:**
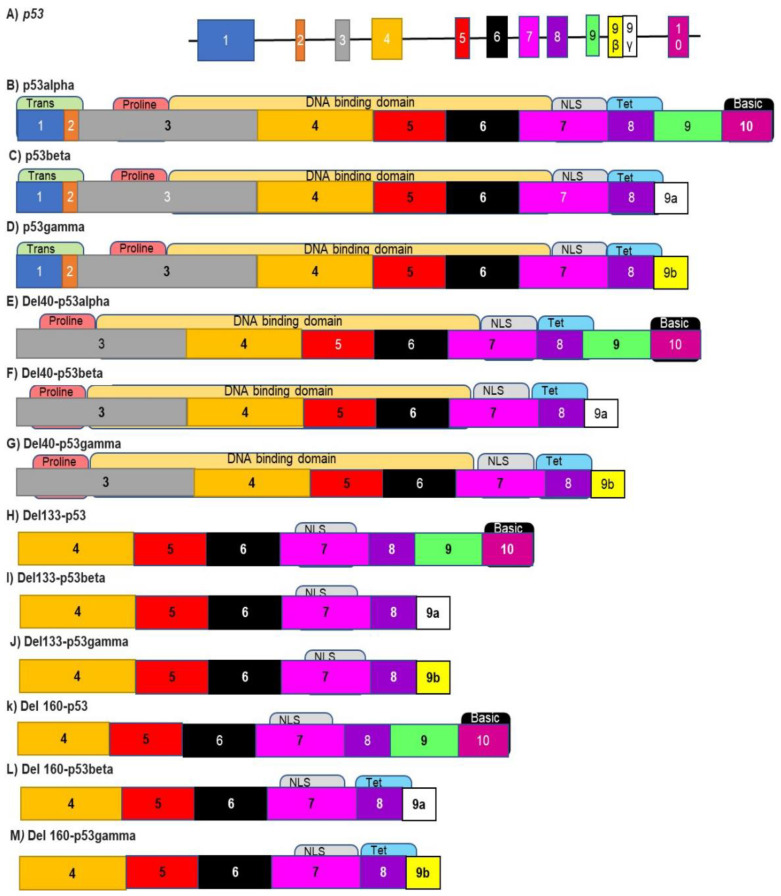
(**A**–**M**) Splicing isoforms of p53: Alternate splicing of p53 gives rise to at least 12 isoforms. The canonical isoforms (**B**) contain all 10 exons and all 6 domains. The main classification of these isoforms relies on differences in the C terminal. The alpha group contains the basic domain encoded by exon 10. The Beta family of isoforms contains the 9a exon, and the gamma exon contains the 9b exon. The extent of the deletions at the N terminal can further divide these into separate groups full length, Del40, Del 133 and Del 160 [64]. The blocks in this figure represent the different exons making up each isoform. The size of the boxes indicates the relative size of the exons.

### 3.5. PITX2

The homeobox gene paired-like homeodomain transcription factor 2 (PITX2) is one of the bicoid transcription factors and is spliced to give rise to four isoforms (PITX2A, PITX2B, PITX2C, PITX2D) (Figure 6). These transcription factors play a role in controlling the transcription of procollagen lysyl hydroxyl, an enzyme responsible for the formation of many body structures during development [65]. PITX2 mutations are responsible for the Axenfeld–Rieger type I condition, a disorder that affects the development of teeth, hair, and abdominal structures [66]. The expression of the PITX2 gene can control the Wnt pathway and interferes with the activation of transcription and the β - catenin cell adhesion mediator. PITX2 is also required for the induction of Cyclins A1 and D2 by recruiting coactivators [67,68]. The different isoforms of PITX2 (A, B and C) induce the transcription of different target genes. Isoform D acts as a negative regulator of the other isoforms by suppressing their transcriptional activity [69]. The isoforms A, B and C are expressed at higher levels in various cancers, where different isoforms stimulate the expression of different members of the TGFβ family [70].

### 3.6. Aberrant Expression of Splicing Factors and Associated Proteins in HNSCC

The expression of HPV proteins is also dependent on the hosts splicing factors. The most important of these splicing factors are hnRNPA1 and hnRNPA2, which control the expression level of the E6 protein, which is directly responsible for the increased levels of HPV-related HNSCC. The viral E7 and E6 proteins are produced due to AS of viral mRNA. The use of the 5’-splice site SD226 and 3’-splice site SA409 produce E7 mRNAs. Un-spliced viral mRNAs produce E6 mRNA, which is promoted by hnRNPA1 by inhibiting the use of the SA409 splice site, decreasing the levels of E7, and increasing those of E6. The levels of E6 are also decreased through splicing induced by hnRNPA2. This splicing factor inhibits the use of the SA-409 splice site and promotes the use of a downstream 3’ splice site named SA742 [71].

## 4. Non-Coding RNAs in HNSCC

Recent studies have indicated that non-coding RNAs (ncRNAs) play an important role in the development and progression of HNSCC. These RNAs regulate the expression of coding genes. MicroRNAs (miRNAs) can either promote or inhibit the expression of target genes by binding directly to their target mRNA. They then affect the stability of the mRNA [72]. This is why the aberrant regulation of miRNAs is an important contributing factor in the development of this disease [73]. LncRNAs may control gene expression by promoting transcription, silencing transcription or by promoting or inhibiting translation [74]. Not only do these ncRNAs regulate the expression of protein-coding genes, but they also regulate the expression of other ncRNAs, and since these molecules act by binding to target mRNA, they also compete for the target binding sites on these mRNA targets. Both these types of ncRNA can also be used to fulfill the role of biomarkers for cancer diagnosis and prognosis, as they are found in the body fluids [75].

### 4.1. MicroRNA Profile in HNSCC

A number of miRNAs have been identified as playing an important role in the development and progression or prevention of HNSCC by acting as either oncogenes or as tumor suppressors [72,76,77,78,79]. AS can generate mRNA with different MicroRNA response elements (MREs) that can alter the ability of miRNA to target them. Different miRNAs can easily be generated through the use of alternate promoters and alternate termination sequences to generate miRNAs with different 5’ and 3’ UTRs. The sequence of miRNAs can also be altered by alternate polyadenylation [80]. An early study that examined miRNA profile changes in HNSCC found that the expression of 20 miRNAs was different in HNSCC samples when compared to normal tissue [76], while a later study using more sensitive deep sequencing found 365 miRNAs with significantly different expression levels in HNSCC samples [81]. Further characterization of these miRNAs that are differentially expressed in HNSCC revealed that 49 of these miRNAs were associated in some way with p53. Sixteen of these miRNAs were also associated with lower survival rates in HNSCC patients [82].

MiRNAs whose expression changes in HNSCC cell lines and patients’ samples that play a tumor suppressor role include miR-200 [83], mi-R375 [84], miR-26a [85], miR-7 [86], miR-107 [87] miR-218 [88] and members of the let-7 micro-RNA family [89]. In addition to this, multiple miRNAs were reported to be downregulated in HNSCC. These include miR-206 [90], miR-10a-5p, miR-125a-5p, miR-144-3p, miR-195-5p and miR-203 [91]. MiR-200 knockdown results in the development of aggressive cancer, while increased levels of, Mi-RNA-200 inhibits cell growth [83]. Another miRNA that acts as a tumor suppressor in multiple cancers, including HNSCC, is mi-R375; however, it was found to act as an oncogene in cancers, such as lung cancer. It was also found that the expression ratio of miR21 to mi-R375 in tumors compared to normal tissue is a good indicator of patient survival. The lower this ratio is, the worse the survival outcome [84]. MiR-26a acts as a tumor suppressor by inhibiting cell migration and metastasis as well as lowering the expression of the enhancer of zeste homolog 2 (EZH2). This results in decreased cell growth [85]. Many of the other tumor suppressor miRNAs function by inhibiting the expression of genes that promote cell proliferation. MiR-7 inhibits EGFR expression [86], miR-107 inhibits Akt, Stat3 and Rho GTPases via Protein kinase Cε (PKCε) [87]. Other tumor suppressor miRNAs function by inhibiting cell migration, invasion, and metastasis by inhibiting signaling cascades. For example, miR-218 inhibits the focal adhesion pathway, preventing cell migration [88].

The changes in miRNA expression in HPV + HNSCC have also been studied. Specific effects of HPV infection in the development of HNSCC rely on the dysregulation of miRNA expression levels and changes in the location of cellular miRNA. MiR-363 is overexpressed in HPV positive HNSCC, where it functions in cell cycle regulation and reduces cell growth and invasion [92,93]. Analysis of the transcription levels of miR-106a and miR-92a did not reveal any variations in expression between HPV+ and HPV- HNSCC cell lines [93], yet in the presence of HPV-16, MiR-155 has been shown to be downregulated [93]. Studies have shown that in HPV+ HNSCC cells, miR-181a and miR-29a were downregulated in comparison to HPV- HNSCC cells [93]. MiR-29a interacts with and stabilizes p53 [94]. Since HPV-16 E6 increases the rate of p53 degradation [20], MiR-29a deregulation in conjunction with E6 expression could further decrease p53 levels following chronic HPV infection [93].

MiRNAs that were found to function as oncogenes include miR-21 [95], miR-375 [96] and miR-184 [97]. Some of the miRNAs that promote HNSCC development and progression that function by the inhibition of apoptosis include miR-21 [95]. Additionally, many of these miRNAs whose expression is increased in HNSCC are also associated with decreased HNSCC survival; an example of this is miR-21 [96]. MiRNAs, who were found to be expressed at higher levels in HNSCC, but whose effects are not known include miR-133b, miR-455-5p and miR-196 [98], miR-26a, miR-21 [95], miR-106b-3p, miR-2, miR-19a, miR-33a and miR-31 [97].

### 4.2. LncRNAs in HNSCC

Multiple studies have identified numerous lncRNAs whose expression is altered in many cancers [97]. As in many other cancers, the lncRNA HOX antisense intergenic RNA (*HOTAIR*) is deregulated in HNSCC. This lncRNA is overexpressed in poorly differentiated HNSCCs, and higher expression is associated with more advanced stages of the disease [99]. Those lncRNAs whose expression is increased in HNSCC include nuclear paraspeckle assembly transcript 1 (NEAT1) [100], HOXA transcript at the distal tip (HOTTIP), urothelial cancer associated 1 (UCA1) [101], lncRNA-regulator of reprogramming (ROR) [102] and H19 [103]. The expression of other lncRNAs is downregulated in HNSCC, and this lower expression is associated with a poorer prognosis. This is a possible indication that they play an antitumor function. These include AC026166.2-001, RP11-169D4.1-001, growth-arrest-specific 5 (GAS5) [100], LET [104], X-inactive specific transcript (XIST) [105], maternally-expressed 3 (MEG3) [106], and lnc-JPHl-7 [107].

## 5. The Contribution of Genomic Mutations to HNSCC

The genomic changes that have been observed in HNSCC include chromosome amplification, chromosome deletion and mutations. Mutations in the genome of HNSCC patients are commonly observed and are known to contribute to cancer development and progression. This has been observed in both HPV+ and HPV- tumors [108]. Some of the most common genes that are found to be mutated in HNSCC include genes that play a role in cell cycle regulation and progression. The gene cyclin-dependent kinase inhibitor 2a (*CDKN2A)* was found to be mutated in up to 87% of HPV- HNSCC tumors. However, mutations in this gene are not common in HPV+ HNSCC [109]. Other groups of genes that are found to be mutated in HNSCC include receptor tyrosine kinases and mitogen-associated protein kinases, growth factors and growth factor receptors [109].

### 5.1. Mutations in P53 and Associated miRNAs

Mutations in TP53 are linked to the poor overall survival of patients when compared to patients with wild-type p53 [110]. Most of the mutations observed in the p53 gene occur in the DNA-binding region, commonly referred to as the mutation hotspot. As a result of this, mutation analysis of the genomic DNA sequence is usually confined to exons 5–8 or 5–9 [111]. A genomic DNA sequence analysis of exons 5–9 was performed in a study of 166 HNSCC patients. Mutations in p53 were found in 65 tumors (39%) [11]. Another study of 32 HNSCC tumors showed that only 8 (25% were identified as having mutations in p53. Once again, this study focused on exons 5–8 [112]. Another later study indicated that in HNSCC, mutations in exons 5–9 were reported in 22% of all tumors [111]. Mutations are also found in the non-coding regions of p53. One mutation is found between exon 6 and 7 (63 bp downstream of exon 6). The resulting mutant protein was found more commonly in samples from cancer patients. It is expected that this mutation results in the stabilization and accumulation of wild-type p53 [113]. Studies estimating the frequency of mutations in p53 in HNSCC samples have found a wide variety of results. These include 78% of HPV- tumors [114]; 60% in freshly frozen HNSCC samples regardless of HPV status [115], to frequencies as low as 30% in some HNSCC samples regardless of HPV infection [108].

In HNSCC, the TP53 tumor suppressor gene is most often mutated in tumors negative for HPV [110]. The tumor suppressor activity of p53 is accomplished through its activity as a transcription factor. Some of the genes whose transcription is initiated by p53 are miRNAs and mRNA coding for proteins involved in cell stress control, apoptosis, and DNA damage repair [110]. One example of this is the expression of miR-377-3p, the primary regulator of Sestrin 1 (SESN1), which encodes a Sestrin family member and is also known as p53-regulated protein PA26 [110]. The expression of SESN1 correlates with the expression of genes that control autophagy. It triggers TP53 expression [110] and helps to stimulate the response to DNA damage and oxidative stress in cells [110]. MiR-377-3p is a downregulator of SESN1, which directly targets 30 untranslated genome regions [110]. The lower expression of miR-377-3p is an indicator of a poor outcome for patients.

### 5.2. Mutation in PIK3CA

The alpha catalytic subunit of phosphatidylinositol-4,5-bisphosphate 3-kinase (*PIK3C*A) has been identified as being the most affected gene in HNSCC. In HPV+ tumors, the mutation hotspots for this gene occur in the area of the gene that codes for the protein’s helical domain [109]. This domain conducts inhibitory signals to the kinase catalytic domain. This kinase activity activates downstream signaling in response to growth factors leading to cell growth. Therefore, mutations that affect the activity of this domain result in uncontrolled growth and cancer [116]. In HPV- tumors, the mutations are not localized to any single region and occur throughout the gene [109].

## 6. Diagnostic and Therapeutic Applications

The differences in mRNA splicing, the resulting changes in expression profiles of various protein isoform and the changes in the transcription levels of ncRNA observed in HNSCC compared to healthy tissue could be used to develop new diagnostic or prognostic biomarkers and may also be utilized as targets for the development of new therapies. There have been multiple clinical trials evaluating alterations in splicing as therapeutic targets, and they have been reviewed in detail elsewhere [117,118].

Drugs have already been developed that target AS in cancer. Some of these drugs are shown in Table 1. One of the therapeutic approaches targeting splicing events for the treatment of cancer is the search for small molecular inhibitors of splicing. This is most commonly done using a bioprospecting approach involving the search for natural products derived from bacteria or plants. Many of the most successful of these compounds inhibit splicing factors, such as SF3B (Table 1). This results in the inhibition of spliceosome assembly [118]. This includes compounds, such as pladienolides, isolated from *Streptomyces platensis*. This compound displays cytotoxic effects and the ability to induce cell cycle arrest. However, this compound is not stable, but a stable derivative molecule was developed based on pladienolide known as E7107 [119]. Although SF3B consists of 7 subunits, named SF3B1, SF3B2, SF3B3, SF3B4, SF3B5, SF3B14, and PHF5A, all these small molecules target SF3B1 [120]. Other than SF3B, many of these compounds inhibit the spliceosome by binding to other snRNPs (Table 1). However, despite the promise of these drugs, cancer cell lines have been shown to develop resistance to them. For instance, after continuous exposure to pladienolide B, human colorectal cell lines developed resistance to the drug. This resistance results from point mutations arising in *SF3B1* (*SF3B1*R1074H), which reduce the binding affinity of these compounds to SF3B [120].

Another means to target specific splicing events is through the use of specific oligonucleotides that are able to hybridize to specific regions of mRNA and by targeting specific sequences and regulate splicing to favor one isoform over another (Table 1) [121]. These oligonucleotides that target specific regions of mRNA need to have the antisense sequence of the mRNA to facilitate binding. Therefore, they have been named antisense oligonucleotides (ASOs). Another type of oligonucleotide-based therapy is known as splice-switching oligonucleotides (SSOs), and many functions by blocking the sites on the mRNA where silencers or enhancers can bind. These exonic splicing enhancer (ESE) sites or intronic splicing silencers (ISSs) can lead to the incorporation of different exons and introns [122]. Despite the promise of these oligonucleotides as therapeutic interventions targeting aberrant splicing, it is important to note that none of these oligonucleotides have been approved by the FDA for the treatment of cancer [121].

Although cancer screening panels do exist that detect splicing-factor mutations, there is as yet none that detect changes in AS events. Additionally, changes in the AS of many genes are shared across many different types of cancers. These changes and the accompanying changes in molecules downstream of these splice variants can be used to provide diagnostic and prognostic biomarkers for many cancers and then be used in specific combinations to stratify individual cancer types. Studies have shown that SESN1 mRNA, UHRF1BP11 mRNA and miRNA-377-3p are important biomarkers for predicting prognosis for patients with HPV- HNSCC [110]. This can help to stratify patients and possibly introduce new clinical strategies for managing HNSCC patients [110]. Further analysis indicated that the mRNA for ubiquitin-like containing PHD and RING finger domains 1-binding protein 1 (UHRF1BP1), and p53-regulated protein PA26 (SESN) are both associated with mutated TP53 [110].

Practical use of these isoform profiles as diagnostic or prognostic markers can be achieved using tissue biopsies from patients. The development of labeled riboprobes specific to the alternately spliced mRNA or labeled antibodies raised against isoform-specific protein regions would allow for the detection of AS variants in these tissue sections. Previously the quantification of the staining intensity in samples, such as these, was based on subjective ranking by an experienced histologist. Recently, automated systems have been developed that can automatically scan and analyze slides prepared in the above manner. An example of such a system is the TissueFAXS (TissueGnostics®, Vienna, Austria). This system has previously been shown to accurately detect the transcription and expression levels of different members of the neuron growth factor (NGF) and the neurotrophin receptor (NTR) families. The expression of different members of the NTR family has been found to be mutually exclusive in different cells, with one member being more commonly expressed in HNSCC (NTRK1, the high-affinity NTR) at higher levels than other members (p75NTR, the low-affinity NTR). Through the use of riboprobes and antibodies specific to these two family members, the TissueFAXS system was able to accurately detect and quantify the levels of these different family members in HNSCC samples [123]. An example of the workflow using such a system is given in Figure 7.

**Table 1 genes-12-00422-t001:** Different classes of drugs that target alternate splicing in cancer.

Class of Compound	Compound	Target Mechanism	Effect on Splicing	
Small molecules	Pladienolides	Abolish the conformation rearrangement of SF3B1	Interfere with canonical splicing cell cycle arrest	[124,125,126]
	Spliceostatins	SF3B1		[127]
	Brr2	U5 snRNP Interfere with the RNA helicase activity	Stall canonical RNA splicing	[128,129]
	Pseudouridine 5 fluorouracil;	U2 snRNA	Induces change in the structure and stability of the branch site, altering base site recognition	
	Sulfonamides	U2AF-related splicing factor	Selective degradation of the U2AF splicing factor	[130]
	Hinokilflavone	U2 snRNA	Blocks spliceosome assembly	[131]
	Jerantimine A	SF3b complex	Inhibits proliferation in human cancer cell lines	[132]
	Amiprophosmethyl	mRNA	mRNA splicing and folding	[132]
Protein inhibitor	TG-003; TG-693	CLK family interfere with ATP binding	Reduced phosphorylation of SRSF family members	[133]
	SRPIN340	SRPK family ATP binding competitor	Altered cellular localization of SRSFs	[134]
	Cpd-1/2/3	SRPK and CLK family	ATP-binding competitor	[122]
Oligonucleotide	ASO-MDM4	MDM4 transcript mRNA degradation	Reduce expression of MDM4 mRNA	[135]
	AZD9150	STAT3 transcript	Reduce expression of STAT3 mRNA	[136]
	AZD4785	KRAS transcript	Reduce expression of KRAS mRNA	[137]

## 7. Conclusions

As the sixth most diagnosed cancer in the world, HNSCC is a large public health burden. The use of changes in the expression profile of alternate protein isoforms or ncRNAs profiles as biomarkers could prove to be a useful diagnostic tool. In addition to this, the isoforms that contribute to cancer development and progression could serve as useful targets for the development of new therapies. These new diagnostic tools and therapeutic targets could assist in the development of personalized healthcare and more precise patient stratification. However, more research is required to establish if these different profiles can truly be used as diagnostic or prognostic tools. The main problem with these markers is that entire profiles of the splicing changes must be evaluated as many different tumors will not show changes in the splicing of the same genes. For example, only a certain percentage of tumors will show alterations in the splicing of one gene, while another percentage of cancer cells will not show any changes in the splicing of the same gene. For an accurate diagnosis and patient stratification, a large amount of data concerning the changes in the populations of various isoforms or mutations will need to be analyzed in order to come to the correct conclusion. This large amount of data could be expensive to obtain for individual patients and may be time-intensive to analyze. As technology advances, this cost will decrease. Additionally, as artificial intelligence and machine learning technology advance, so the time needed to analyze data will decrease.

These molecular differences between healthy and HNSCC tissue allow patients to be defined by their prognosis and optimize the management of the disease through specific treatments or, in the worst circumstances, palliative care. Multiple methods of targeting alternative splicing are being developed. However, problems have been encountered with many of these approaches. Small molecules that can target the spliceosome, inhibiting the formation of specific splice variants, can lead to resistance to these drugs. However, the development of new drugs derived from naturally occurring compounds that can inhibit splicing may provide novel therapies that cancer cells do cannot readily develop resistance to. Despite the hurdles and the large amount of further study required to develop new biomarkers and therapeutic targets based on genetic drivers of HNSCC. These molecular profiles still offer a promising target for improving treatment outcomes and creating new diagnostic and prognostic tests for HNSCC.

## Figures and Tables

**Figure 1 genes-12-00422-f001:**
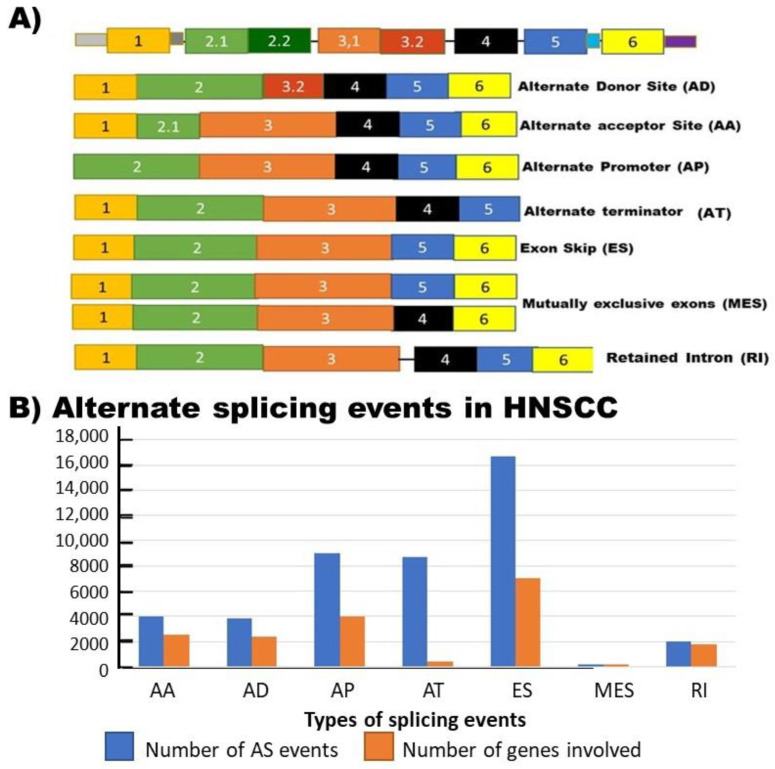
Analysis of alternative splicing (AS) events in head and neck squamous cell carcinomas (HNSCC). (**A**) Seven types of AS events were identified in HNSCC patient samples. These include changes in the location of alternate acceptor sites (AA), donor sites (AD), alternate promoter sites (AP) and alternate terminator sites (AT). Other AS events involve changes in the incorporation of exons and exclusion of introns due to exon skipping (ES), the use of mutually exclusive exons (ME) and retained introns (RI) [40]. (**B**) The number of each type of AS event taking place in 519 HNSCC patients. AA, alternative accepter; AD, alternative donor; AP, alternative promoter. AT, alternative terminator site; ES, exon skip; ME, mutually exclusive exons; RI, intron preserved/retained intron [39].

**Figure 2 genes-12-00422-f002:**
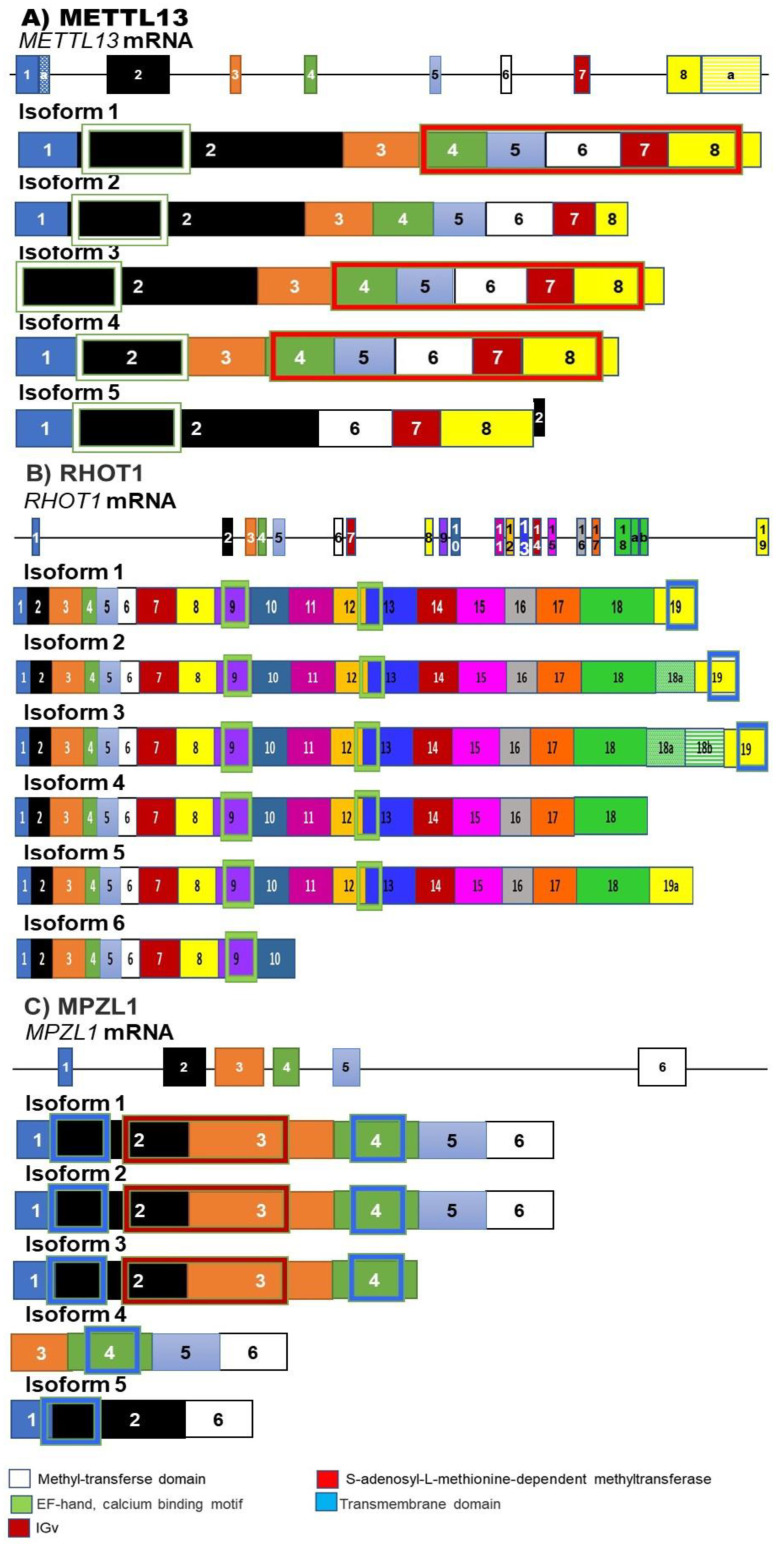
Isoforms of some of the proteins where alternative splicing had a strong correlation with survival. (**A**) eEF1A lysine and N-terminal methyltransferase (METTL13) are responsible for the activation of the eukaryotic elongation factor 1A (eEF1A) through methylation. There are 5 known isoforms of this protein. Some of these isoforms are missing one of the two methyltransferase domains. (**B**) Ras homolog gene family member T1 (RHOT1) is the gene that codes for the Mitochondrial Rho GTPase protein, a membrane receptor that is spliced to give rise to at least 6 isoforms. Some of these isoforms lack the transmembrane receptor, implying that these isoforms can block signaling. (**C**) Myelin protein zero-like protein 1 (MPZL1) is spliced to give rise to 5 known isoforms, some of which lack one of the transmembrane domains. Numbered boxes indicate exons, while colored boxes show the position of the corresponding domain. The blocks in this figure represent the different exons making up each isoform. The size of the boxes indicates the relative size of the exons.

**Figure 3 genes-12-00422-f003:**
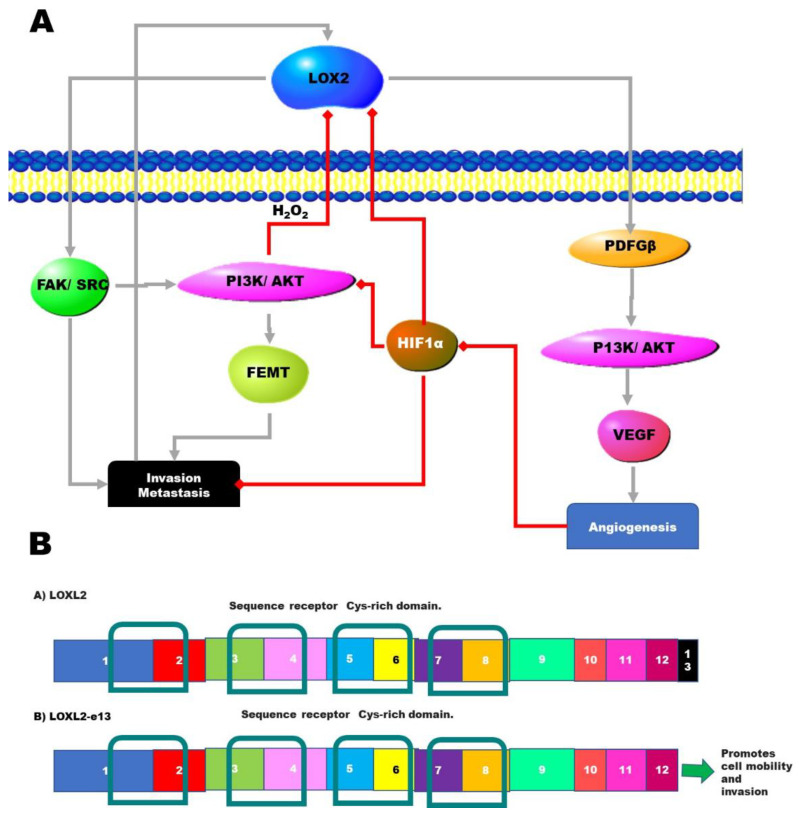
Alternate splicing of the LOXL2 mRNA. (**A**) The lysyl oxidase-like (LOXL2) signaling pathway stimulates invasion and angiogenesis in cancer cells. The pathway is also inhibited because of hypoxia. Green lines with green arrows indicate stimulation or induction, while red lines with red diamonds indicate inhibition. (**B**) The e13 splice variant of LOXL2 results from the exclusion of the final exon, exon 13. This variant promotes stronger signaling promoting invasion and metastasis. Both isoforms contain all four SR domains (scavenger receptor Cys-rich domain). These are shown in the figure as the blue boxes. These domains are responsible for facilitating binding to cell membranes during phagocytosis. FAK, focal adhesion kinase; PI3K/AKT, phosphatidylinositol 3-kinase/protein kinase B; HIF1α, hypoxia-inducible factor 1α; VEGF, vascular endothelial growth factor; EMT, epithelial-mesenchymal transition.

**Figure 4 genes-12-00422-f004:**
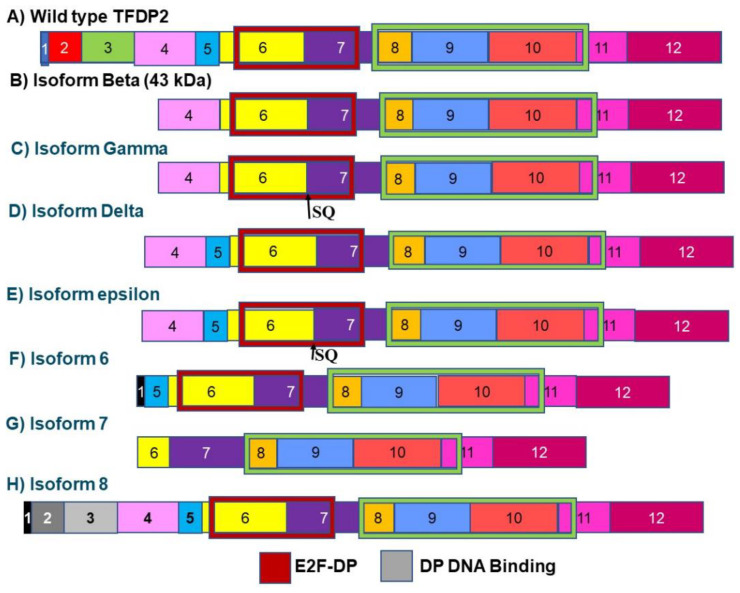
(**A**–**H**) Isoforms of TFDP2. The TFDP2 transcription factor is known to be expressed at high levels in HNSCC and is alternately spliced to give rise to 8 isoforms. Isoform 7 is missing the E2F/DP dimerization domain. This isoform can block E2F associated transcription and, therefore, inhibit cell cycle progression. Although the precise role of these isoforms in HNSCC is unknown, the high expression levels of the wild-type variant in HNSCC imply that isoform 7 may serve as a negative regulator of cancer progression. The blocks in this figure represent the different exons making up each isoform. The size of the boxes indicates the relative size of the exons.

**Figure 6 genes-12-00422-f006:**
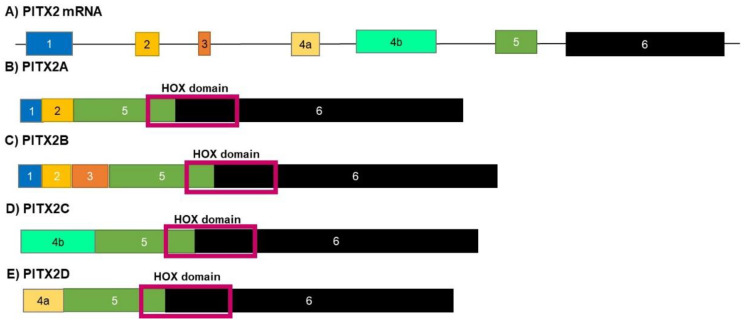
Isoforms of PITX2. Alternate splicing of the PITX2 mRNA gives rise to four protein isoforms. Three of these isoforms (**A**–**C**) have a similar function, each inducing the transcription of different genes that stimulate growth and proliferation. The final isoform, isoform (**D**), acts as a negative regulator of the other three isoforms. The blocks in this figure represent the different exons making up each isoform. The size of the boxes indicates the relative size of the exons (**E**).

**Figure 7 genes-12-00422-f007:**
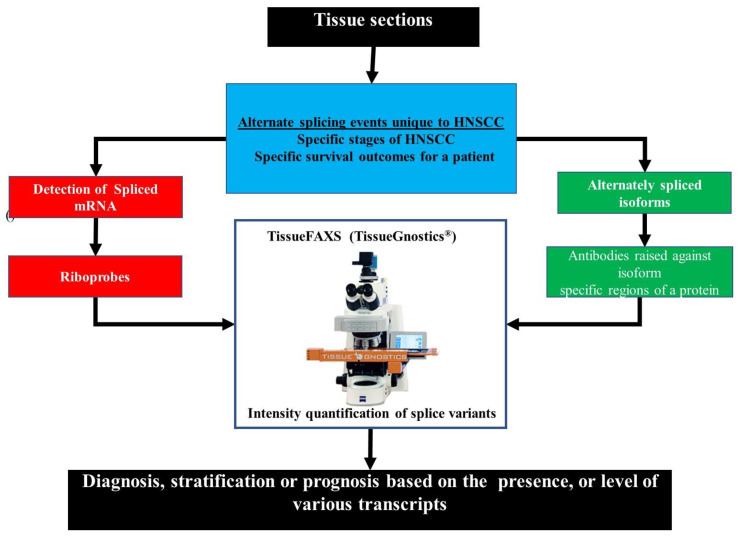
An example of a workflow to detect alternative splicing events using an automated detection and quantification system. The differences in the profiles of splicing events can be used to diagnose HNSCC, detect if the HNSCC is related to human papillomavirus (HPV) infection or stratify patients based on cancer stage and severity. Either riboprobes specific for mRNA transcripts or antibodies that are raised against portions of different protein isoforms that are unique to specific isoforms can be used to detect variants. An automated quantification system, such as the TissueFAXS Plus from TissueGnostics, can be used to rapidly and accurately detect and quantify the levels of these variants without the possible inaccuracies introduced by a subjective assessment by a histologist.

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
