# Peer review of "Genetic Drivers of Head and Neck Squamous Cell Carcinoma: Aberrant Splicing Events, Mutational Burden, HPV Infection and Future Targets"

_genes, 2021, doi:10.3390/genes12030422_

Round 1
Reviewer 1 Report
Dilami et al present a review on recent advances in Head and Neck Squamous Cell Carcinoma.
The quality of the writing is sometimes quite flaw and not very scientific. The novelty/improvements are not immediately evident.
The epigenetic approach is not very evident, thus the justification to mention it in the title is quite questionable.
There are treatment options already, but they are not well explained.
I would suggest shortening some parts of the manuscript which are quite "scholarly" especially in the introduction.
In the conclusions, no real opinion or perspective is given despite "future perspectives" are mentioned in the title.
Some figures need to be improved for visual quality. Some References have annotations in the text, but no actual reference in the relevant section
The topic of diagnostic and therapeutic applications is very flaw. For example Table 1 has no appropriate explanation in the text. The various approaches should be explained better and not only summarized in a table. Furthermore novel innovative concepts should be explained and cited.
Author Response
1)The quality of the writing is sometimes quite flaw and not very scientific.
Response:
We have tried to the best of our abilities to improve the writing of this manuscript and tried to make it to be more scientific
2) The novelty/improvements are not immediately evident.
Response:
We also improved this and also taking into consideration concerns of
reviewer. These additionas include expanded therapeutics and conclusions
sections.
3) The epigenetic approach is not very evident, thus the justification to mention it in the title is quite questionable.
Response:
The term epigenetic drivers has been removed from the title. References to epigenetic drivers of HNSCC are only mentioned briefly throughout the manuscript. This is often in reference to methylation of genes or proteins. As such we have decided that it is not correct to indicate that it is a major theme of the review.
4)There are treatment options already, but they are not well explained.
Response
These treatment options are now discussed in more detail in section 6.
5) I would suggest shortening some parts of the manuscript which are quite "scholarly" especially in the introduction.
Response:
Many sections including the introduction have been shortened.
6) In the conclusions, no real opinion or perspective is given despite "future perspectives" are mentioned in the title.
Response
Opinions and perspectives are now discussed in the conclusion.
7) Some figures need to be improved for visual quality.
Response
Figures have been improved and certain details have been enlarged to improve visual quality.
8) Some References have annotations in the text, but no actual reference in the relevant section.
Response:
All references now have a corresponding entry in the reference section.
9) The topic of diagnostic and therapeutic applications is very flaw. For example Table 1 has no appropriate explanation in the text.
Response:
This section has been improved, with a discussion concerning the relevant therapeutic approaches. Reference is made in these additions to Table 1.
10) The various approaches should be explained better and not only summarized in a table.
Response:
This section has been improved, with a discussion concerning the relevant therapeutic approaches. Reference is made in these additions to Table 1.
11) Furthermore novel innovative concepts should be explained and cited.
Response:
These concepts are now explained with citations.
Reviewer 2 Report
Comments to the Authors
In this review article, Dlamini et al. summarize and discuss the most common genetic and epigenetic alterations that drive the initiation and progression of head and neck squamous cell carcinoma (HNSCC). More specifically, they address the most critical alternative splicing events, gene mutations, and the role of HPV infection. The review is thorough and contains interesting information, detailing useful data regarding HNSCC, and molecules with biomarker and therapeutic potential. The Figures and Table are helpful for the reader’s comprehension. However, there are several issues in the Manuscript that need to be addressed accordingly, before this manuscript becomes acceptable for publication in Genes:
Major issues:
- In the Introduction it is stated that methylation is a genomic change; however, it is an epigenetic modification. Moreover, the way the period is written is confusing. Do you refer to DNA methylation, protein methylation, or both? In any case, this type of modification cannot be found by genome-wide sequencing, therefore this period needs to change in order to be clearer and more precise.
- It would be beneficial to discuss whether any molecules mentioned in this review could be used as non-invasive biomarkers, for liquid biopsies.
- Circular RNAs, another type of RNA molecules, have recently been in the spotlight due to their unique features and multilayered implication in cancer development, progression, and resistance to therapy. In my opinion, it would be interesting and worthwhile to include some information regarding the role of circular RNAs in the context of HNSCC.
- What are some limitations of the use of genetic and epigenetic drivers mentioned in the Manuscript, as therapy targets? I believe it would be interesting to discuss potential problems and restrictions for clinical application.
- The Conclusion serves as the final impression of the Manuscript, that includes the major points of the review. Therefore, new information should be avoided in this part, such as the use of antisense oligonucleotides. This information is better suited in the previous paragraph.
- Regarding all figures that depict alternatively spliced mRNAs, does the length of each box correspond to the actual relative length in nucleotides of each exon? Or is the box length chosen randomly?
Minor issues:
- What do the blue boxes in Figure 3B exactly point out? Please specify in the respective figure legend.
- The full name of the HPV should be included in the Abstract.
- In the Introduction, the term “long non-coded RNAs” should be replaced by the correct term “long non-coding RNAs”.
- The term used in the Abstract “…alternately spliced protein isoforms” is perhaps misleading since splicing occurs in premature RNA molecules and not in the protein. Authors should consider rephrasing this period.
- In paragraph 5. PITX2, Authors mention that “Changes in biopsy samples which require decisions to be taken before surgery or radio chemotherapy could be of value in clinical practice”. Please clarify what you mean, since the point of this statement and the relation to previous information is unclear.
- The acronym ‘UTR’ should also be explained the first time it is used in the Manuscript.
- There should be cohesion and consistency in names and terminology throughout the text. For example, all abbreviations and acronyms should be mentioned only once, the first time a term is mentioned; every additional use is redundant.
- In paragraph Diagnostic and Therapeutic Applications, there is a reference that does not follow the citation style of the manuscript: {Dudás, 2018 #226}.
- English language used in the manuscript needs improvement, since there are several grammatical, syntax and typing errors throughout the Abstract and Manuscript. Authors are advised to carefully proof-read the text, possibly with the help of a native English speaker.
Author Response
Major issues:
- In the Introduction it is stated that methylation is a genomic change; however, it is an epigenetic modification. Moreover, the way the period is written is confusing. Do you refer to DNA methylation, protein methylation, or both? In any case, this type of modification cannot be found by genome-wide sequencing, therefore this period needs to change in order to be clearer and more precise.
Response:
This statement has been corrected and clarified. It is now made clear that the sequencing data was used to predict possible methylation sites.
- It would be beneficial to discuss whether any molecules mentioned in this review could be used as non-invasive biomarkers, for liquid biopsies.
Response:
This has been included in the section for LOXL2, which is secreted and can therefore be used as a biomarker in liquid biopsies.
- Circular RNAs, another type of RNA molecules, have recently been in the spotlight due to their unique features and multilayered implication in cancer development, progression, and resistance to therapy. In my opinion, it would be interesting and worthwhile to include some information regarding the role of circular RNAs in the context of HNSCC.
Response:
These ncRNAs are recently implicated in cancer development and progression. There is very little information related to HNSCC. Additionally, many of the comments over this manuscript involve the shortening of the manuscript. As such we believe it is not feasible to include a section with such a small amount of information available
- What are some limitations of the use of genetic and epigenetic drivers mentioned in the Manuscript, as therapy targets? I believe it would be interesting to discuss potential problems and restrictions for clinical application.
Response:
Some disadvantages of some of these therapeutic approaches is now discussed while describing the various therapies.
- The Conclusion serves as the final impression of the Manuscript, that includes the major points of the review. Therefore, new information should be avoided in this part, such as the use of antisense oligonucleotides. This information is better suited in the previous paragraph.
Response:
No new information is included in the conclusion. The problematic statement has been removed from the conclusion.
- Regarding all figures that depict alternatively spliced mRNAs, does the length of each box correspond to the actual relative length in nucleotides of each exon? Or is the box length chosen randomly?
Response:
The size of the boxes does correspond to the relative size of the exons. This is now explained in the legends of these figures.
Minor issues:
- What do the blue boxes in Figure 3B exactly point out? Please specify in the respective figure legend.
Response:
This is now explained in the figure legend as representing the SR domains.
- The full name of the HPV should be included in the Abstract.
Response:
The full name is now given in the abstract.
- In the Introduction, the term “long noncoding RNAs” should be replaced by the correct term “long non-coding RNAs”.
Response:
This has been corrected throughout the paper.
- The term used in the Abstract “…alternately spliced protein isoforms” is perhaps misleading since splicing occurs in premature RNA molecules and not in the protein. Authors should consider rephrasing this period.
Response:
This statement has been corrected.
- In paragraph 5. PITX2, Authors mention that “Changes in biopsy samples which require decisions to be taken before surgery or radio chemotherapy could be of value in clinical practice”. Please clarify what you mean, since the point of this statement and the relation to previous information is unclear.
Response:
This statement is unnecessary and has been removed.
- The acronym ‘UTR’ should also be explained the first time it is used in the
Manuscript.
Response:
This is now done the first time UTR is mentioned in section 3.5.
7. There should be cohesion and consistency in names and terminology throughout the text. For example, all abbreviations and acronyms should be mentioned only once, the first time a term is mentioned; every additional use is redundant.
Response:
The manuscript has been revised and the use of abbreviations and italicised, gene names has been standardised.
8. In paragraph Diagnostic and Therapeutic Applications, there is a reference that does not follow the citation style of the manuscript: {Dudás, 2018 #226}.
Response:
The reference has been inserted.
9. English language used in the manuscript needs improvement, since there are several grammatical, syntax and typing errors throughout the Abstract and Manuscript. Authors are advised to carefully proof-read the text, possibly with the help of a native English speaker.
Response:
The Paper has been revised and spell checked to improve the English.
Round 2
Reviewer 2 Report
The manuscript has been substantially improved and is now suitable for publication in Genes.